# Benchmarking CXR Foundation Models With Publicly Available MIMIC-CXR and NIH-CXR14 Datasets

## Abstract

Recent foundation models have demonstrated strong performance in medical image representation learning, yet their comparative behaviour across datasets remains underexplored. This work benchmarks two large-scale chest X-ray (CXR) embedding models (**CXR-Foundation** (**ELIXR v2.0**) and **MedImageInsight**) on public **MIMIC-CXR** and **NIH ChestX-ray14** datasets. Each model was evaluated using a unified preprocessing pipeline and fixed downstream classifiers to ensure reproducible comparison. We extracted embeddings directly from pre-trained encoders, trained lightweight LightGBM classifiers on multiple disease labels, and reported mean AUROC, and F1-score with 95% confidence intervals. MedImageInsight achieved slightly higher performance across most tasks, while CXR-Foundation exhibited strong cross-dataset stability. Unsupervised clustering of MedImageInsight embeddings further revealed a coherent disease-specific structure consistent with quantitative results. The results highlight the need for standardised evaluation of medical foundation models and establish reproducible baselines for future multimodal and clinical integration studies.

## 1 Introduction

Recent advances in large-scale chest X-ray (CXR) representation learning have led to the development of foundation and embedding models that map high-dimensional radiological data into compact feature spaces with strong generalisation capabilities. Studies such as CheXzero [Tiu et al., 2022], BioViL [Boecking et al., 2022], and CXR-CLIP [Nguyen et al., 2022] demonstrated that self-supervised and vision–language pretraining can achieve radiologist-level performance on multi-label classification and zero-shot transfer tasks. Such embeddings can facilitate large-scale cohort analysis and support patient subtyping when combined with structured clinical data.

The **MIMIC-CXR** database [Johnson et al., 2019] and the **NIH ChestX-ray14** dataset [Wang et al., 2017] are the most widely used public datasets for benchmarking medical image models. Despite rapid progress in vision-language pretraining, few systematic comparisons have been performed between recent foundation encoders on these datasets. In particular, the performance gap between Google's **CXR-Foundation** model [Google-Research, 2023] and Microsoft's **MedImageInsight** model [Microsoft-Research, 2024] remains underexplored.

This study provides a reproducible benchmark of these two embedding models across the MIMIC-CXR and NIH ChestX-ray14 datasets. Using identical preprocessing and downstream classifiers, we evaluate their ability to represent clinically meaningful image variations across common thoracic disease labels. The results establish reference points for researchers aiming to integrate CXR embeddings into multimodal or clinical decision-support pipelines.

## 2 Methods

### 2.1 Datasets

We used two public chest radiography datasets: **MIMIC-CXR** (377k images from 227k studies) and **NIH ChestX-ray14** (112k frontal images from 30k patients) [Johnson et al., 2019, Wang et al., 2017]. Only frontal (PA/AP) projections were retained; lateral views and corrupted files were excluded. For each disease label (Atelectasis, Edema, Effusion, Opacity), we sampled 1,000 positive and 1,000 negative images from each dataset (MIMIC-CXR and NIH-CXR14). All splits were made by unique patient-ids: 80% training and 20% test for classification tasks.

### 2.2 Preprocessing

Images were read using `pydicom`, rescaled using manufacturer metadata, converted to MONOCHROME2, and normalised to $[0, 1]$. They were resized to $1024 \times 1024$ for CXR-Foundation inference and standardised by z-score normalisation. To probe representational stability, five augmented views per training image were generated: two small rotations ($\pm 5$–$10°$), two brightness shifts ($\pm 10$–$15\%$), and one contrast scaling ($\pm 10\%$).

### 2.3 Embedding Models

Two pretrained vision–language encoders were evaluated: **CXR-Foundation (ELIXR v2.0)** and **MedImageInsight**. Each CXR produced token features $\mathbf{t}_i$ that were mean-pooled into a single embedding. All embeddings were stored as 32x768 (CXR-Foundation) or 1024 (MedImageInsight) vectors.

### 2.4 Dimensionality Reduction and Clustering

We used Uniform Manifold Approximation and Projection (UMAP) for visualisation [McInnes et al., 2018] and applied **k-means** clustering (cosine distance, $n_{init}$=50) implemented in scikit-learn [Pedregosa et al., 2011]. The optimal cluster number $k$ maximised the mean Silhouette coefficient

$$\mathcal{S} = \frac{1}{N} \sum_{i=1}^{N} \frac{b_i - a_i}{\max(a_i, b_i)}, \tag{1}$$

where $a_i$ and $b_i$ are intra- and inter-cluster distances.

### 2.5 Evaluation

To gauge representational quality, frozen embeddings were used to train lightweight **LightGBM** classifiers [Ke et al., 2017] on selected pathology labels using 5-fold patient-wise cross-validation. Performance was summarised by mean AUROC, and F1 with 95 % confidence intervals.

### 2.6 Reproducibility

Experiments were run in Python 3.10 on a single NVIDIA A100 (40 GB) GPU using open-source packages (`pydicom`, `scikit-learn`, `umap-learn`, `lightgbm`). All datasets are publicly available and fully de-identified [Goldberger et al., 2000].

## 3 Results

Table 1 summarises the performance of MedImageInsight and CXR-Foundation across four thoracic disease labels from MIMICand NIH. Both models achieved strong performance, with mean AUROC values above 0.90 for most tasks. MedImageInsight generally outperformed CXR-Foundation, showing higher AUROC and F1-scores across most labels. Performance trends were consistent across datasets, indicating that both embedding spaces generalise well between domains.

UMAP projections of MedImageInsight embeddings (Figure 1) reflect this pattern: Effusion shows distinct separation between positive and negative samples, while Opacity exhibits more overlap,

Table 1: Benchmark of **MedImageInsight** vs. **CXR-Foundation** on MIMIC-CXR and NIH ChestX-ray14. Values are mean ± 95% CI.

| Task | | AUROC | | F1 | |
|---|---|---|---|---|---|
| Disease | Dataset | MedImageInsight | CXR-Foundation | MedImageInsight | CXR-Foundation |
| Atelectasis | MIMIC | $0.833 \pm 0.007$ | $0.823 \pm 0.013$ | $0.755 \pm 0.007$ | $0.751 \pm 0.008$ |
| | NIH | $0.863 \pm 0.008$ | $0.822 \pm 0.012$ | $0.782 \pm 0.015$ | $0.744 \pm 0.014$ |
| Edema | MIMIC | $0.918 \pm 0.011$ | $0.924 \pm 0.014$ | $0.841 \pm 0.014$ | $0.847 \pm 0.014$ |
| | NIH | $0.921 \pm 0.012$ | $0.911 \pm 0.006$ | $0.853 \pm 0.016$ | $0.831 \pm 0.013$ |
| Effusion | MIMIC | $0.958 \pm 0.011$ | $0.941 \pm 0.014$ | $0.906 \pm 0.013$ | $0.877 \pm 0.010$ |
| | NIH | $0.901 \pm 0.012$ | $0.901 \pm 0.006$ | $0.828 \pm 0.014$ | $0.826 \pm 0.008$ |
| Opacity | MIMIC | $0.782 \pm 0.019$ | $0.775 \pm 0.017$ | $0.702 \pm 0.016$ | $0.704 \pm 0.023$ |
| | NIH | $0.922 \pm 0.012$ | $0.955 \pm 0.006$ | $0.851 \pm 0.019$ | $0.889 \pm 0.013$ |

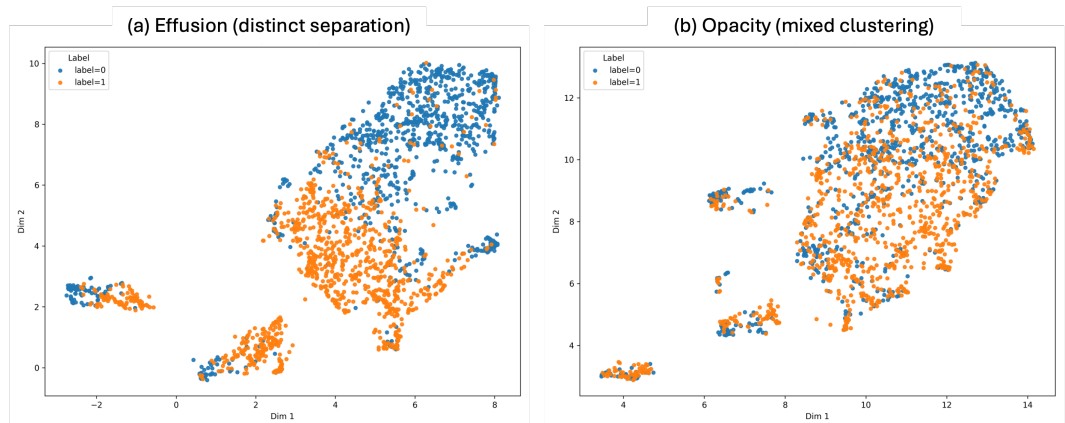

Figure 1: UMAP visualisation of MedImageInsight embeddings for the highest- and lowest-performing disease labels on MIMIC dataset: (a) **Effusion** shows distinct separation between positive and negative samples, consistent with its high AUROC, while (b) **Opacity** displays mixed clustering, reflecting lower discriminative power. 0 indicates absence and 1 indicates presence of disease label.

consistent with the quantitative results. These findings demonstrate the robustness of MedImageInsight representations and establish reproducible baselines for future multimodal fusion or clinical stratification studies.

## 4 Discussion and Limitations

MedImageInsight generally outperformed CXR-Foundation across thoracic disease labels, suggesting that compact, well-aligned representations enhance model stability and generalisation. Its 1024-dimensional embedding strikes a balance between expressiveness and efficiency, beneficial in multimodal or large-scale settings where computation and memory are limited. Prior studies show that reducing embedding dimensionality can improve regularisation and cross-modal alignment [Baltrusaitis et al., 2019, Tsai et al., 2019, Chen et al., 2023].

Clustering analysis of MedImageInsight embeddings revealed coherent latent structures across pathologies, indicating that its representations capture meaningful visual distinctions. This organised embedding space aligns with prior work on self-supervised radiograph learning [Boecking et al., 2022, Nguyen et al., 2022, Tiu et al., 2022] and suggests strong potential for future multimodal fusion, patient subtyping, and interpretable feature analysis.

**Limitations.** This study examined two foundation models using frontal chest X-rays and unsupervised clustering. Results may differ with alternative architectures, fine-tuning, or lateral views.

## 5  Potential Negative Societal Impact.

Although MIMIC-CXR and NIH ChestX-ray14 are de-identified, they reflect limited demographic and institutional diversity. Models trained or benchmarked on such datasets may inherit hidden biases or perform inconsistently across underrepresented groups. This work is intended purely for methodological benchmarking and not for clinical application. Openly sharing such analyses encourages transparency and critical evaluation of foundation models in medical imaging.

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
