# OpenReview forum: "Benchmarking CXR Foundation Models With Publicly Available MIMIC-CXR and NIH-CXR14 Datasets"
_EurIPS.cc/2025/Workshop/MedEurIPS — EurIPS 2025 Workshop MedEurIPS Submission_

### Official Review · Reviewer_5mcK · 2025-10-24
**Limited novelty compared to existing work**

**Rating:** 5
**Confidence:** 5

**Review:**

This paper presents an evaluation study comparing two CXR foundation models (CXR-Foundation and MedImageInsight) on two publicly available datasets (MIMIC-CXR and NIH-CXR-14). Although the empirical comparison can be valuable, I believe that the contribution of this work is quite limited. More specifically, the original papers for these models already report results on these datasets (see [1] for CXR-Foundation and [2] for MedImageInsight). Most importantly, while the authors claim that MedImageInsight outperforms CXR-Foundation in this benchmark, the results presented in Table 1 do not appear statistically significant. I recommend that the authors (1) evaluate these models on different datasets, ideally on out-of-distribution data with respect to each model’s training set, and (2) clarify what novel insights this evaluation provides beyond existing work.

[1] Xu, Shawn, et al. "Elixr: Towards a general purpose x-ray artificial intelligence system through alignment of large language models and radiology vision encoders." arXiv preprint arXiv:2308.01317 (2023).
[2] Codella, Noel CF, et al. "Medimageinsight: An open-source embedding model for general domain medical imaging." arXiv preprint arXiv:2410.06542 (2024).

---

### Official Review · Reviewer_wBsc · 2025-10-31
**Review- Benchmarking CXR Foundation Models With Publicly Available MIMIC-CXR and NIH-CXR14 Datasets**

**Rating:** 4
**Confidence:** 4

**Review:**

The paper compares the representational differences of two foundation models on chest X-ray data.

Strengths:
- The motivation to gather generalizable insights for the research community is good.
- The structure of the paper is clear, and the results are well presented.
- The method compares multiple metrics and multiple datasets.

Weaknesses and Suggestions:
- The presentation of the paper could be improved. The space is not used optimally, and the paper provides unnecessary details about well-known methods (e.g., dimensionality reduction), making  overly
- In Figure 1, it would be interesting to see whether the mislabeled samples cluster.
- The diseases selected for evaluation appear to be arbitrarily chosen. It would be better to evaluate all labels or at least clarify why only two of the four classes were shown.
- There is no comparison to naive methods. Since the datasets were heavily subsampled, it would be beneficial to include a comparison to simple fine-tuning or foundation models trained on natural images.
- Predictive performance was evaluated using LightGBM, but it is unclear how the results are affected by the large differences in feature extractor size. It would be useful to test other learning-based evaluation methods, such as linear probing, to better assess representational quality.
- The results do not generalize well. Since the evaluation was conducted on only a subset of the datasets and labels, the overall generalizability remains unclear.

---

### Decision · Program_Chairs · 2025-10-31

**Decision:**

Reject

**Comment:**

Both reviewers acknowledge that the benchmarking study is clearly structured and reproducible, but they find the novelty limited. The comparison replicates evaluations already presented in prior works and lacks broader or out-of-distribution analyses.